# Toward Efficient Influence Function: Dropout as a Compression Tool

**Yuchen Zhang**  *zhangy94@rpi.edu*
*Department of Computer Science*
*Rensselaer Polytechnic Institute*

**Mohammad Mohammadi Amiri**  *mohamm11@rpi.edu*
*Department of Computer Science*
*Rensselaer Polytechnic Institute*

**Reviewed on OpenReview:** *https://openreview.net/forum?id=rapeA5Ha3C*

## Abstract

Assessing the impact the training data on machine learning models is crucial for understanding the behavior of the model, enhancing the transparency, and selecting training data. Influence function provides a theoretical framework for quantifying the effect of training data points on model's performance given a specific test data. However, the computational and memory costs of influence function presents significant challenges, especially for large-scale models, even when using approximation methods, since the gradients involved in computation are as large as the model itself. In this work, we introduce a novel approach that leverages **dropout** as a gradient compression mechanism to compute the influence function more efficiently. Our method significantly reduces computational and memory overhead, not only during the influence function computation but also in gradient compression process. Through theoretical analysis and empirical validation, we demonstrate that our method could preserves critical components of the data influence and enables its application to modern large-scale models.

## 1 Introduction

Large foundation models such as GPT-4 (Achiam et al., 2023), Llama (Grattafiori et al., 2024), and DeepSeek (Liu et al., 2024), have showcased remarkable capabilities across a variety of tasks. Despite their success, even the state-of-the-art models face persistent challenges, including hallucination (Lin et al., 2021; Huang et al., 2025) and the generation of toxic or biased content (Abid et al., 2021; Wang et al., 2023). A critical factor underlying these shortcomings is the composition and quality of their training data (Park et al., 2023). Furthermore, training data also impart the knowledge that forms the foundation of a model's capabilities (Wang et al., 2024; Meng et al., 2022; Mirzadeh et al., 2024). This raises a critical question: which data contribute positively to a model's performance, and which ones negatively impact it? Addressing this highlights the need for robust methods to evaluate the influence of training data.

Influence function, a theoretical method rooted in statistics (Hampel, 1974; Law, 1986), which was originally used to assess the robustness of statistical estimator (Huber & Ronchetti, 2011), provides a powerful tool for assessing the impact of training data on a model's parameters and subsequently on the model's performance. It offers a framework to understand how modifications to the training dataset propagate through the model. The concept has since been adapted to deep learning (Koh & Liang, 2017; Koh et al., 2019), enabling its application to modern large-scale models. This method has been wildly used in training data selection (Xia et al., 2024; Yu et al., 2024; Hu et al., 2024), data synthesizing (Li et al., 2024), and mislabel data detection (Koh & Liang, 2017; Kwon et al., 2023).

Although the influence function provides a robust framework and has demonstrated promising results, their practical application is often hampered by high computational costs (Kwon et al., 2023; Zhou et al., 2024; Choe et al., 2024). Computing influence function involves calculating an inverse Hessian-vector product (iHVP) and the gradients of the loss function with respect to both training and testing data. Since the Hessian matrix's dimensionality scales quadratically with the size of the model, and each gradient is as large as the model itself, these processes become prohibitively expensive for large-scale model, both in terms of computation and memory. Previous methods have attempted to mitigate the overhead of influence function through approximation or compression techniques. Approximation methods typically rely on iteration (Agarwal et al., 2017; Koh & Liang, 2017) or closed form approximation (Kwon et al., 2023), but the large size of gradients poses a fundamental challenge, even storing all gradients can be impractical for large-scale models. Compression methods, such as those using random Gaussian projections (Park et al., 2023) or principal component analysis (PCA) (Choe et al., 2024), alleviate the challenge posed by the large size of gradients, but they introduce additional memory and computational overhead during the compression process, as shown in Figure 1. These approaches still face challenges in scaling to modern large-scale models or adapting to diverse model architectures and structure, which further limits their practicality.

In this work, we focus on compression-based approaches to influence estimation. Research has shown that modern machine learning (ML) models are highly overparameterized (Balaji et al., 2021; Fischer et al., 2024), with only a small subset of parameters playing a critical role in their performance (Fedus et al., 2022; Xue et al., 2024). Furthermore, previous studies indicate that the influence of training data is closely tied to the high spectrum of the Hessian matrix, where the majority of eigenvalues are concentrated near zero, and only a few outliers deviate significantly from the bulk (Sagun et al., 2016; 2017). These findings highlight that tracking data influence does not require exhaustive computation over the entire parameter space, but can focus on a few critical directions or a small subset of parameters.

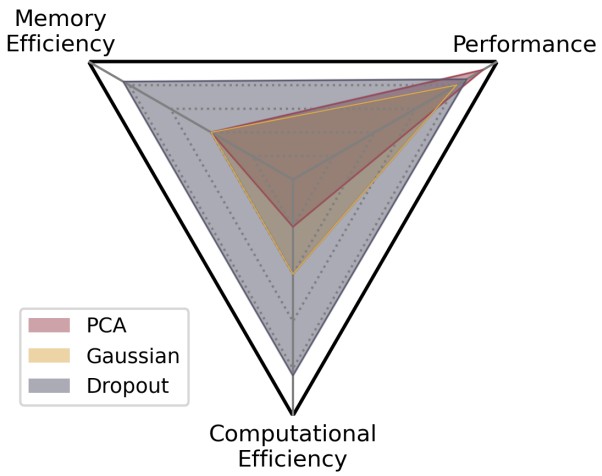

Figure 1: Comparison of different compression methods for influence function estimation. PCA identifies important directions but incurs high computational overhead. Both PCA and Gaussian projection require storing a compression map, which can be memory intensive. In contrast, Dropout avoids both computational and memory overhead, making it a more efficient alternative.

**Our Contributions.** We observe that the influence of training data on the performance of a ML model can be effectively tracked through a small subset of parameters, reducing the need to consider the full parameter space. Building on this, we propose a novel dropout-based compression method to compress gradients, which is both straightforward to implement and scales efficiently to large-scale ML models. This significantly reduces both memory and computational complexity associated with computation of influence function and the compression process itself. Through theoretical analysis and empirical experiments, we validate the effectiveness of the proposed method, demonstrating its ability to capture data influence while offering efficiency.

## 2 Preliminaries

We denote the input space and the output space by $\mathcal{X}$ and $\mathcal{Y}$, respectively. Let $\mathcal{D}_{\mathrm{tr}} = \{z_{\mathrm{tr}}^1, z_{\mathrm{tr}}^2, \cdots, z_{\mathrm{tr}}^n\}$ represent the training dataset, where each training data point $z_{\mathrm{tr}}^i = (x_{\mathrm{tr}}^i, y_{\mathrm{tr}}^i) \in \mathcal{X} \times \mathcal{Y}$. For a given data point $z = (x, y)$ and a model with parameters $\theta \in \Theta$, let $l(y, f_\theta(x))$ denote the loss function, where $f_\theta : \mathcal{X} \to \mathcal{Y}$ is the model parameterized by $\theta$, and $l : \mathcal{Y} \times \mathcal{Y} \to \mathbb{R}$ measures the discrepancy between the output and

the ground truth. The gradient of loss function evaluated at the data point $z$ with respect to $\theta$ is denoted as $\nabla_\theta l\left(y, f_\theta(x)\right)$. Additionally, let $\mathcal{D}_{\text{val}} = \{z_{\text{val}}^1, z_{\text{val}}^2, \cdots, z_{\text{val}}^m\}$ denote the validation dataset, where each validation data point $z_{\text{val}}^j = (x_{\text{val}}^j, y_{\text{val}}^j) \in \mathcal{X} \times \mathcal{Y}$. Finally, we denote the number of parameters in the model by $d$.

## 2.1 Influence Function

The influence function quantifies how the model parameters change in response to upweighting a specific training data point, and how the change affects the model's performance (Hampel, 1974; Law, 1986; Koh & Liang, 2017). Formally, given an infinitesimally small $\epsilon > 0$, the upweighted empirical risk minimization problem is formulated by increasing the weight of the $k$-th training data point $z_{\text{tr}}^k = (x_{\text{tr}}^k, y_{\text{tr}}^k)$ in the loss function. The optimization problem is given by:

$$\theta^{(k)}(\epsilon) = \arg\min_{\theta \in \Theta} \frac{1}{n} \sum_{i=1}^n l\left(y_{\text{tr}}^i, f_\theta(x_{\text{tr}}^i)\right) + \epsilon l\left(y_{\text{tr}}^k, f_\theta(x_{\text{tr}}^k)\right).$$

Assuming the loss function is twice-differentiable and strongly convex in $\theta$, the influence of the $k$-th training data point on the empirical risk minimizer $\theta^*$ is defined as the derivative of $\theta^{(k)}(\epsilon)$ at $\epsilon = 0$ (Koh & Liang, 2017):

$$\mathcal{I}_{\theta^*}(z_{\text{tr}}^k) := \left.\frac{\mathrm{d}\theta^{(k)}(\epsilon)}{\mathrm{d}\epsilon}\right|_{\epsilon=0} = -\boldsymbol{H}^{-1}\boldsymbol{g}_{\text{tr}}^k,$$

where $\boldsymbol{H} := \frac{1}{n}\sum_{i=1}^n \nabla_\theta^2 l\left(y_{\text{tr}}^i, f_\theta(x_{\text{tr}}^i)\right)\big|_{\theta=\theta^*}$ is the empirical Hessian matrix and $\boldsymbol{g}_{\text{tr}}^k = \nabla_\theta l\left(y_{\text{tr}}^k, f_\theta(x_{\text{tr}}^k)\right)\big|_{\theta=\theta^*}$ represents the gradient of the loss function evaluated at the $k$-th training data point $z_{\text{tr}}^k$.

For the validation dataset $\mathcal{D}_{\text{val}} = \{z_{\text{val}}^1, z_{\text{val}}^2, \cdots, z_{\text{val}}^m\}$, the influence of the training data point $z_{\text{tr}}^k$ on the validation loss is (Koh & Liang, 2017; Kwon et al., 2023):

$$\mathcal{I}(z_{\text{tr}}^k) := \frac{1}{m}\sum_{j=1}^m \left(\boldsymbol{g}_{\text{val}}^j\right)^\top \mathcal{I}_{\theta^*}(z_{\text{tr}}^k) = -\frac{1}{m}\sum_{j=1}^m \left(\boldsymbol{g}_{\text{val}}^j\right)^\top \boldsymbol{H}^{-1}\boldsymbol{g}_{\text{tr}}^k, \tag{1}$$

where $\boldsymbol{g}_{\text{val}}^j = \nabla_\theta l\left(y_{\text{val}}^j, f_\theta(x_{\text{val}}^j)\right)\big|_{\theta=\theta*}$ is the gradient of loss function evaluated at $z_{\text{val}}^j$.

The influence function $\mathcal{I}(z_{\text{tr}}^k)$ provides an intuitive method to evaluate whether a training data point $z_{\text{tr}}^k$ is beneficial or detrimental to the performance of the model on the validation dataset $\mathcal{D}_{\text{val}}$. When the loss function is cross-entropy loss, the Hessian matrix could be approximated with the Fisher-Information Matrix (FIM), which is equivalent to the Gauss-Newton Hessian (Martens, 2020; Bae et al., 2022; Grosse et al., 2023). Note that $\boldsymbol{H}$ is not invertible if the dimension of $\theta$ exceeds the size of training dataset $n$, which is common in many modern ML models. To address the issue, a damping term is added to $\boldsymbol{H}$, i.e., replacing $\boldsymbol{H}$ with $\boldsymbol{H} + \lambda\boldsymbol{I}_d$, where $\lambda$ is a small constant, and $\boldsymbol{I}_d$ is a $d \times d$ identity matrix.

## 2.2 Compressing Gradients for Influence Function

Computing the influence function faces several challenges when $f_\theta$ is a large-scale deep learning model (Basu et al., 2020; Bae et al., 2022; Kwon et al., 2023). A key obstacle is that the size of the Hessian becomes prohibitively large to compute directly, as its dimensionality scales quadratically with the number of the model parameters.

To address this challenge, several methods (Schioppa et al., 2022; Park et al., 2023) propose projecting gradients onto a low-dimensional subspace using a random Gaussian projection matrix (Johnson et al.,

1984) and computing the influence function in the subspace as follows:

$$\tilde{\mathcal{I}}_{\text{Gaussian}}(z_{\text{tr}}^k) = -\frac{1}{m} \sum_{j=1}^{m} \left(\boldsymbol{g}_{\text{val}}^j\right)^{\top} \boldsymbol{P}^{\top} \left(\frac{1}{n} \sum_{i=1}^{n} \boldsymbol{P} \boldsymbol{g}_{\text{tr}}^i \boldsymbol{g}_{\text{tr}}^{i\top} \boldsymbol{P}^{\top}\right)^{-1} \boldsymbol{P} \boldsymbol{g}_{\text{tr}}^k$$

$$= -\frac{1}{m} \sum_{j=1}^{m} \left(\boldsymbol{g}_{\text{val}}^j\right)^{\top} \boldsymbol{P}^{\top} (\boldsymbol{P} \boldsymbol{H} \boldsymbol{P}^{\top})^{-1} \boldsymbol{P} \boldsymbol{g}_{\text{tr}}^k, \tag{2}$$

where $\boldsymbol{P} \in \mathbb{R}^{r \times d}$ is a random Gaussian projection matrix. Here, $r$ represents the dimensionality of the compressed subspace and $r \ll d$. While influence function computation becomes more efficient in terms of computation and memory complexity, the use of a projection matrix $\boldsymbol{P}$ introduces additional computing and memory overhead for compression (Choe et al., 2024). Specifically, computing a gradient via backpropagation has a cost of $O(d)$, whereas projecting it into a lower-dimensional subspace using $\boldsymbol{P}$ incurs a cost of $O(rd)$. Even efficient projection methods, such as FJLT (Ailon & Chazelle, 2009) have a higher computational complexity than $O(d)$, making the compression process more expensive than the gradient computation itself. Additionally, storing these compression maps can incur memory overhead which exceeds the memory usage of the model itself. These limitations highlight the need for a new compression strategy for influence function computation, which minimizes both computing and memory costs, while preserving key information necessary for reliable influence estimation.

## 3 Method

To address the computational and memory challenges associated with influence function computation, we propose a novel approach that leverages dropout as a gradient compression mechanism. We demonstrate that the influence of training data on a small subset of parameters can effectively reflect its influence on the entire parameter space. Unlike traditional compression methods, which require a random Gaussian matrix as the compression map or use PCA to obtain the important components, incurring significant memory and computational costs, our method simply drops a random subset of gradient entries. As shown in Figure 1, this technique reduces the dimensionality of gradients without incurring the additional memory and computational overhead typically associated with the compression process.

### 3.1 Dropout as a Compression Mechanism

Dropout is a widely used regularization technique in deep learning (Srivastava et al., 2014), where a subset of model parameters or activations is randomly set to zero during training. We apply a similar approach to the gradient vectors during influence function computation, compressing the gradient by retaining only a small subset of its entries. Let $\boldsymbol{g} \in \mathbb{R}^d$ represent the gradient of the loss function with respect to the model parameters for a data $z = (x, y)$, i.e. $\boldsymbol{g} = \nabla_\theta l\left(y, f_\theta(x)\right)\Big|_{\theta=\theta^*}$. To compress the gradient, we randomly sample $r$ entries of the gradient $\boldsymbol{g}$. Mathematically, this process is equivalent to using a binary matrix $\tilde{\boldsymbol{I}} \in \mathbb{R}^{r \times d}$ to compress the gradient $\boldsymbol{g}$ and get the compressed one $\tilde{\boldsymbol{g}}$:

$$\tilde{\boldsymbol{g}} = \tilde{\boldsymbol{I}} \boldsymbol{g} \in \mathbb{R}^r.$$

Each row of $\tilde{\boldsymbol{I}}$ has exactly one entry equal to 1, while all other entries are 0, and there is only one non-zero entry in each column:

$$\tilde{\boldsymbol{I}} \in \{0,1\}^{r \times d}, \quad \sum_{j=0}^{d-1} \tilde{\boldsymbol{I}}_{ij} = 1 \quad \forall i \in \{0, 1, \ldots, r-1\}. \tag{3}$$

Specifically, $\tilde{\boldsymbol{I}}_{ij} = 1$ indicates that the $j$-th entry of the gradient is retained.

It is important to note that the introduction of $\tilde{\boldsymbol{I}}$ is used for theoretical analysis, and in practice we do not need to explicitly construct this. This avoids unnecessary computational and memory overhead, thereby simplifying the implementation while maintaining efficiency. For a detailed implementation pipeline and step-by-step algorithmic description, please refer to Algorithm 1 in Appendix A.2.

After getting compressed gradients $\tilde{\boldsymbol{g}}$, we use compressed gradients for influence function computation:

$$
\begin{aligned}
\tilde{\mathcal{I}}_{\text{Dropout}}(z_{\text{tr}}^k) &= -\frac{1}{m} \sum_{j=1}^m (\tilde{\boldsymbol{g}}_{\text{val}}^j)^\top \tilde{\boldsymbol{H}}^{-1} \tilde{\boldsymbol{g}}_{\text{tr}}^k \\
&= -\frac{1}{m} \sum_{j=1}^m (\tilde{\boldsymbol{I}} \boldsymbol{g}_{\text{val}}^j)^\top (\tilde{\boldsymbol{I}} \boldsymbol{H} \tilde{\boldsymbol{I}}^\top)^{-1} \tilde{\boldsymbol{I}} \boldsymbol{g}_{\text{tr}}^k.
\end{aligned}
\tag{4}
$$

The matrix $\tilde{\boldsymbol{H}}$ is the Hessian/Gauss-Newton Hessian calculated using compressed gradients.

### 3.2 Efficiency Comparison

Even though traditional gradient compression methods, such as random projection (Johnson et al., 1984) used in TRAK (Park et al., 2023), reduce the complexity of computation of influence function, they rely on explicit projection matrices to compress. That will introduce significant memory and computational overhead, because these methods use dense projection matrices with a memory complexity $O(rd)$ and computational complexity dominated by matrix-vector multiplication, which is $O(rd)$ for each gradient. PCA requires even more resources to obtain the principal components. In contrast, our dropout compression method avoids the need for explicit projection matrices, and reduce memory and computational costs to $O(r)$ as only $r$ entries of gradients are sampled and stored.

Other efficient influence function computation methods, such as LiSSA (Agarwal et al., 2017) and LOGRA (Choe et al., 2024), employ stochastic iterative approaches or Kronecker product for gradient computation, respectively. While these methods reduce the computational cost of iHVP, they still require expensive iterative algorithm (Klochkov & Liu, 2024) or are hard to expand to all deep learning architectures (Kwon et al., 2023; Grosse et al., 2023). The comparison of computational and memory costs of different methods for gradient compression and influence function computation are detailed in Appendix A.3.

### 3.3 Error Analysis

While the dropout-based compression method in equation 4 offers a more efficient alternative for computing influence functions than the standard Gaussian approach equation 2, it is important to understand the trade-offs in accuracy. Intuitively, Gaussian projects compresses gradient information into a lower-dimensional subspace, whereas dropout retains only a random subset of gradient entries, potentially discarding significant information. To address this, we theoretically analyze the upper bound of the error incurred by our method. For both methods, the compression error is defined as the difference $\mathcal{I}(z_{tr}^k) - \tilde{\mathcal{I}}_{\text{Gaussian or Dropout}}(z_{tr}^k)$. The spectral norm of this error is expressed as:

$$
\begin{aligned}
\left\| \mathcal{I}(z_{\text{tr}}^k) - \tilde{\mathcal{I}}_{\text{Gaussian or Dropout}}(z_{\text{tr}}^k) \right\|_2 &= \left\| \left( \frac{1}{m} \sum_{j=1}^m \boldsymbol{g}_{\text{val}}^j \right)^\top \Delta \boldsymbol{H} \boldsymbol{g}_{\text{tr}}^k \right\|_2 \\
&\leq \left\| \frac{1}{m} \sum_{j=1}^m \boldsymbol{g}_{\text{val}}^j \right\|_2 \left\| \Delta \boldsymbol{H} \right\|_2 \left\| \boldsymbol{g}_{\text{tr}}^k \right\|_2,
\end{aligned}
\tag{5}
$$

and it is governed by the term $\|\Delta \boldsymbol{H}\|_2$, where $\Delta \boldsymbol{H}$ represents the difference between the full inverse Hessian and its compressed counterpart.

Our theoretical analysis indicates that the error introduced by Gaussian-based compression can actually have a higher theoretical upper bound compared to dropout-based approach. Specifically, Gaussian-based compression error is bounded by $O\left(d + d^2 \sigma_{\max}(\boldsymbol{H})\right)$, while dropout-based compression error is bounded by $O\left(\sigma_{\max}(\boldsymbol{H})\right)$. These results provide a worst-case stability guarantee, ensuring that the approximation remains robust even at high compression rates. We provide the formal statements in Appendix A.4. This theoretical framework supports the utility of dropout as a lightweight and practical tool for influence estimation, particularly when computational resources are constrained.

Table 1: Performance of mislabeled data detection on some GLUE benchmarks (MRPC, QNLI, QQP, SST2) using various methods for influence function computing. The reported results are averaged over 5 independent runs, with standard deviations shown as subscripts. The best results (excluding `Orig`) are highlighted in **bold**, and the second-best results are underlined.

| Method | Rank=2 | | | | Rank=8 | | | |
|---|---|---|---|---|---|---|---|---|
| | MRPC | QNLI | QQP | SST2 | MRPC | QNLI | QQP | SST2 |
| `Orig` | $0.832_{0.005}$ | $0.794_{0.071}$ | $0.807_{0.015}$ | $0.802_{0.028}$ | - | - | - | - |
| `LiSSA` | $0.665_{0.018}$ | $0.498_{0.014}$ | $0.578_{0.079}$ | $0.509_{0.056}$ | $0.617_{0.075}$ | $0.500_{0.015}$ | $0.609_{0.046}$ | $0.478_{0.046}$ |
| `Hessian-free` | $0.687_{0.011}$ | $0.639_{0.128}$ | $0.666_{0.046}$ | $0.814_{0.116}$ | $0.691_{0.009}$ | $0.697_{0.132}$ | $0.627_{0.016}$ | $0.610_{0.184}$ |
| `DataInf` | $0.789_{0.013}$ | $0.753_{0.144}$ | $0.723_{0.076}$ | $\underline{0.934}_{0.006}$ | $0.794_{0.028}$ | $0.793_{0.100}$ | $0.785_{0.059}$ | $0.821_{0.193}$ |
| `Gaussian` | $0.823_{0.012}$ | $0.786_{0.124}$ | $\underline{0.793}_{0.034}$ | $0.933_{0.005}$ | $\underline{0.844}_{0.026}$ | $0.833_{0.082}$ | $0.821_{0.046}$ | $\underline{0.828}_{0.206}$ |
| `PCA` | $\mathbf{0.833}_{0.013}$ | $\mathbf{0.792}_{0.125}$ | $\mathbf{0.803}_{0.030}$ | $0.932_{0.007}$ | $\mathbf{0.850}_{0.024}$ | $\mathbf{0.841}_{0.061}$ | $\mathbf{0.832}_{0.036}$ | $0.827_{0.203}$ |
| `Dropout`(Ours) | $\underline{0.825}_{0.010}$ | $\underline{0.791}_{0.037}$ | $0.791_{0.037}$ | $\mathbf{0.935}_{0.007}$ | $0.842_{0.025}$ | $\underline{0.836}_{0.073}$ | $\underline{0.825}_{0.045}$ | $\mathbf{0.838}_{0.188}$ |

## 4 Experiments

In this section, we evaluate the effectiveness of our method: Using dropout as a compression tool for influence function computation, in terms of accuracy and efficiency, key factors in real-world data attribution tasks. We conduct three experiments to evaluate the effectiveness of our approach: (1) mislabeled data detection 4.1, which uses influence function to identify mislabeled data points in a noisy training dataset; (2) model retraining 4.2, which identifies the most influential training examples and retrains the model either with only these data points or with them removed to observe their impact on model performance; and (3) cross-source influential data identification 4.3, which investigates whether influential training examples typically originate from the same source as their corresponding test examples. To comprehensively evaluate our method, we start with relatively small experimental setups and then scale up to billion-parameter models. This allows us to assess how well our method generalizes across settings and to demonstrate its scalability. Additional experimental details are provided in Appendix A.5.

### 4.1 Mislabeled Data Detection

Mislabeled data points often negatively impact a model's performance. It is expected that the influence values of these mislabeled data points will be larger than clean data points, as their unweighting tends to increase the testing loss.

In this experiment, we use five binary classification datasets from GLUE benchmark (Wang et al., 2018), and synthetically generate mislabeled training data points similar to (Kwon et al., 2023), flipping the binary label for 20% of randomly selected training data points to simulate the situation where a part of data points are noisy. We use the RoBERTa model (Liu et al., 2019) and fine-tune the model on those noisy dataset using LoRA (Hu et al., 2022) with 2-rank and 8-rank separately. As for the baselines, we investigate the performance of five efficient methods, including three approximation methods and two compression methods, as well as the `Orig` influence function in equation 1. For approximation methods, we consider `LiSSA` (Koh & Liang, 2017) with 10 iterations, `Hessian-free` which only computes the dot product of gradients (Pruthi et al., 2020), `DataInf` which uses an closed form approximation version of influence function (Kwon et al., 2023). For compression methods, we consider `Gaussian`, which uses a random Gaussian matrix to compress gradients similar to (Park et al., 2023), and `PCA` that uses PCA to obtain principal components to compress gradients. Some details of these methods are attached in Appendix A.1. For compression methods, including `Gaussian`, `PCA`, and `Dropout`, we use $r = 16$ for both 2-rank and 8-rank LoRA.

For evaluation metrics, we use the area under the curve (AUC). The AUC quantifies the ability of the influence function to distinguish between mislabeled and clean data points. Specifically, it measures the probability that a score selected from a class of mislabeled data is greater than that of a class of clean data.

An influence function that reliably assigns larger influence values to mislabeled data points will achieve a high AUC score, reflecting its effectiveness in identify mislabeled data.

**Results.** Table 1 compares the mislabeled data detection performance of various influence function computation methods using LoRA with ranks 2 and 8. Results are averaged over 5 independent runs. `Dropout` achieves comparable, and in some cases superior detection performance compared to `Gaussian` and `PCA`, despite requiring no additional computation or memory overhead for compression. Furthermore, there is a consistent trend indicating that compression-based methods outperform approximation-based ones. Similar to findings in Kwon et al. (2023), we observe that original uncompressed gradients do not always have the best performance (`Orig`). This is potentially because the whole gradients contain some redundant information which has negative impacts on the performance. In terms of running time, compression based methods demonstrate superior computational efficiency for iHVP computation. For example, on QNLI dataset with 8-rank LoRA, compression methods take an average of 4.83 seconds, whereas `DataInf` takes 13.38 seconds. More results on time usage of iHVP computation of these methods across various benchmarks is provided in Appendix A.5.2.

## 4.2 Model Retraining

The retraining process begins by identifying the most influential data points. The new training dataset constructed by these influential data points or removing these highly influential data points from the original dataset. The performance of the retrained model is evaluated on the original test dataset. Retraining could demonstrate the critical role of these influential data in model's learning process and the effectiveness of methods used to identify these influential data.

### 4.2.1 Small-Scale Setups

We initiate this experiment with small-scale setups: (1) ResNet-9 (He et al., 2016) with CIFAR-10, in which we train a ResNet-9 model from scratch using a randomly selected subset containing 10000 data points and evaluate on a test dataset containing 256 data points by accuracy. Then we remove a specific amount of influential data from the training dataset and retrain the model. Large performance decrease indicates greater effectiveness of the method in identifying the influential data. (2) GPT-2 (Radford et al., 2019) with CNN daily mail, in which we full fine tune a GPT-2 using 1000 text samples and evaluate on a test dataset containing 512 text samples by perplexity. Then we only use a specific amount of influential data to retrain the model. Large perplexity decrease indicates greater effectiveness of the method in identifying the influential data. We use influence function to compute the influential score of each training data and rank them by these score. On these benchmarks, we compare `Dropout` against baselines include: `Random Ranking`, which randomly rank training data; `LiSSA` (Koh & Liang, 2017) uses an iteration method to get influential scores; `DataInf` (Kwon et al., 2023) uses a form of approximated influence function; `Hessian-free` (Pruthi et al., 2020) computes the dot product of gradients directly; `LOGRA` (Choe et al., 2024) uses Kronecker product for gradients computation and compression; `Gaussian` uses random Gaussian matrix to compression gradients; `FJLT`, an efficient compression method proposed in Ailon & Chazelle (2009); and `PCA` uses PCA to obtain principal components and do compression. Some details of these methods are attached in Appendix A.1. For `LOGRA` we use $r_{\text{in}} = r_{\text{out}} = \sqrt{r} = 64$, and for `Gaussian`, `PCA` and `Dropout` we use $r = 64$ to do compression.

**Results.** The retraining results of ResNet-9 and GPT-2 are in Table 2 and Table 3, separately. We observe that `Dropout` achieved performance comparable to or better than other methods across both settings. `LOGRA` achieves strong performance in ResNet-9, it does not perform as well in GPT-2. We hypothesize that this is due to the Kronecker structure used in `LOGRA`, which may not generalize well across architectures. In terms of efficiency, compression methods demonstrate impressive performance in computing iHVP same as previous. Moreover, the efficiency of gradients compression becomes more crucial when dealing with large-scale models. Notably, the `Dropout` excels in efficiency during the compression process. For example, in GPT-2 experiment, `FJLT` uses an average of 453.87 seconds to compress all gradients, in contrast, `Dropout` requires only 9.76 seconds, representing a 46× speedup in the compression process. Additionally, `Dropout` eliminates the extra memory overhead associated with storing the full compressing map, which is a requirement for

Table 2: Accuracy (%) of ResNet-9 on the test dataset after removing a specific amount of the most influential training data from the training dataset. The reported results are averaged over 5 independent runs, with standard deviations shown as subscripts. The testing accuracy of the model trained on the full training dataset is 78.83%. The best results are highlighted in **bold**, and the second-best results are underlined.

| Methods | 5% | 15% | 25% | 35% |
|---|---|---|---|---|
| Random Ranking | $79.14_{1.07}$ | $76.68_{2.39}$ | $75.66_{0.42}$ | $74.06_{0.42}$ |
| LiSSA | $78.01_{1.38}$ | $75.35_{1.79}$ | $69.73_{2.84}$ | $65.16_{1.48}$ |
| Hessian-free | $78.01_{1.22}$ | $74.30_{0.97}$ | $70.27_{0.81}$ | $65.51_{1.66}$ |
| DataInf | $77.73_{0.39}$ | $73.28_{0.83}$ | $69.30_{1.29}$ | $63.91_{0.45}$ |
| LOGRA | $\mathbf{76.09}_{0.74}$ | $\mathbf{71.33}_{1.53}$ | $\mathbf{67.97}_{1.22}$ | $64.30_{0.87}$ |
| Gaussian | $77.81_{1.18}$ | $72.73_{1.59}$ | $\underline{68.40}_{1.49}$ | $63.32_{0.67}$ |
| PCA | $77.62_{1.16}$ | $72.34_{1.40}$ | $69.06_{1.21}$ | $\underline{63.63}_{0.76}$ |
| Dropout(Ours) | $\underline{76.88}_{0.76}$ | $\underline{72.27}_{1.48}$ | $68.87_{1.62}$ | $\mathbf{62.30}_{0.57}$ |

Table 3: Testing perplexity of GPT-2 trained with a specific amount of the most influential training data. The reported results are averaged over 5 independent runs, with standard deviations shown as subscripts. The testing perplexity of the model trained on the full dataset is 13.67. The best results are highlighted in **bold**, and the second-best results are underlined.

| Methods | 5% | 10% | 15% | 20% |
|---|---|---|---|---|
| Random Ranking | $17.43_{0.20}$ | $17.38_{0.07}$ | $17.10_{0.04}$ | $17.05_{0.05}$ |
| LiSSA | $17.40_{0.11}$ | $17.31_{0.08}$ | $17.11_{0.06}$ | $17.05_{0.07}$ |
| Hessian-free | $17.43_{0.26}$ | $17.13_{0.05}$ | $\mathbf{16.91}_{0.03}$ | $16.90_{0.08}$ |
| DataInf | $18.04_{0.62}$ | $\underline{17.11}_{0.06}$ | $\mathbf{16.91}_{0.02}$ | $\underline{16.87}_{0.05}$ |
| LOGRA | $17.65_{0.43}$ | $17.24_{0.17}$ | $17.03_{0.13}$ | $16.89_{0.08}$ |
| FJLT | $\mathbf{17.25}_{0.06}$ | $17.14_{0.05}$ | $16.98_{0.10}$ | $16.91_{0.03}$ |
| Dropout(Ours) | $\underline{17.27}_{0.12}$ | $\mathbf{17.10}_{0.09}$ | $\underline{16.95}_{0.08}$ | $\mathbf{16.84}_{0.05}$ |

compression methods like `Gaussian`, `FJLT`, `PCA` and `LOGRA`. The time consumption of these methods is detailed in Appendix A.5.2.

### 4.2.2 Large-Scale Settings

We now evaluate our approach for data attribution in billion-parameter models, using Pythia-1.4B and Pythia-6.9B (Biderman et al., 2023). For Pythia-1.4B, we perform data attribution on a subset of OpenWeb-Text (OWT) (Gokaslan et al., 2019) with 2,000 training and 200 testing text sample. For Pythia-6.9B, we use a heterogeneous dataset with six sources, including CNN daily mail, math reasoning, wikisql, java code, and others, using 10,700 training and 300 testing samples.

In both cases, we fully fine-tune the model and then remove the top-$k$ percent most influential data from the training dataset and retrain the model. A larger performance drop indicates a more effective method for data attribution. Notably, the gradients used in the computation of influence function is the same size as the model itself. This makes most gradient-based methods, including `Hessian-free`, impractical for billion-scale models due to the prohibitive resources required to store $O(d)$ per-example gradients. Therefore, we evaluate only the methods feasible under our setup: `Random Ranking`, `LOGRA` with $r_{\text{in}} = r_{\text{out}} = \sqrt{r} = 64$, and `Dropout` with $r = 128$.

**Results.** Table 4 presents the retraining results for Pythia 1.4B and Table 5 for Pythia 6.9B. We should realize that when the training dataset becomes large, the marginal contribution of each individual data point diminishes. As a result, removing a small subset of data has only a marginal impact on the model's

Table 4: Testing perplexity of Pythia 1.4B on test dataset after removing a specific amount of the most influential data from the training dataset. The reported results are averaged over 5 independent runs. The testing perplexity of the model trained on the full training dataset is 25.23. The best results are highlighted in **bold**, and the second-best results are underlined.

| Methods | 10% | 20% | 35% | 40% |
|---|---|---|---|---|
| Random Ranking | 25.52 | 25.91 | 26.28 | 26.49 |
| LOGRA | **25.68** | 25.89 | **26.62** | 26.79 |
| Dropout(Ours) | 25.51 | **26.05** | 26.55 | **26.80** |

Table 5: Testing perplexity of Pythia 6.9B on test dataset after removing a specific amount of the most influential data from the training dataset. The reported results are averaged over 5 independent runs. The testing perplexity of the model trained on the full training dataset is 2.54. The best results are highlighted in **bold**.

| Methods | 10% | 20% | 35% | 40% |
|---|---|---|---|---|
| Random Ranking | 2.54 | 2.55 | 2.55 | 2.56 |
| Dropout(Ours) | **2.55** | **2.56** | **2.57** | **2.58** |

performance when fine-tuning Pythia 6.9B on over ten thousands of text samples. Nonetheless, our method still shows slightly better results under these conditions.

### 4.3 Cross-Source Influential Data Identification

To assess whether our method can effectively identify the influential training examples in large-scale settings, we apply the influence function with `Dropout` compression to examine whether the identified most influential samples for training originate from the same class or data source as their corresponding validation examples. We conduct this experiment using the Pythia-6.9B model, fine-tuned on a dataset composed of six heterogeneous sources, including CNN daily mail, math reasoning, wikisql, and others, containing 10,700 training and 300 testing samples, consistent with the previous setup. Importantly, our method remains computationally feasible under this setup, as it avoids the need to save full per-example gradients or large compression matrices, making it both memory and compute efficient.

**Results.** Intuitively, for a large model like Pythia trained on heterogeneous data sources (such as CNN Daily Mail, Math Reasoning, WikiSQL, and others), a successful data attribution method should be able to trace a given test sample back to training samples that originate from the same source. To quantify this, we measure the source alignment using Top-1 and Top-3 accuracy. Top-1 calculates the proportion of test samples where the most influential training sample originates from the same source. Top-3 is a stricter metric that calculates the proportion of test samples where all three of the most influential training samples originate from the same source. Given our six-source dataset, the Top-1 and Top-3 accuracy for a `Random Ranking` baseline (randomly selecting training samples) is 16.7% and 0.7%, separately. Our method's performance (see Table 6) substantially exceeds this, confirming that dropout preserves essential semantic signals even at high compression ratios. Table 6 reports the results and examples of identified influential training data are shown in Appendix A.5.3

## 5 Analysis

In this section, we provide an additional error analysis to complement the results in Section 3.3. Our theoretical results in Section 3.3 indicated that the spectral norm of the error for `Dropout` possesses a tighter upper bound compared to that of the `Gaussian` approach. While these bounds offer a worst-case stability guarantee, they do not necessarily reflect the expected error or the variance of the estimates across different

Table 6: Proportion of **top-1** and **top-3** most influential training examples that belong to the same class as the corresponding test example.

| Methods | top-1 | top-3 |
|---|---|---|
| Random Ranking | 0.167 | 0.007 |
| Dropout(Ours) | 0.831 | 0.642 |

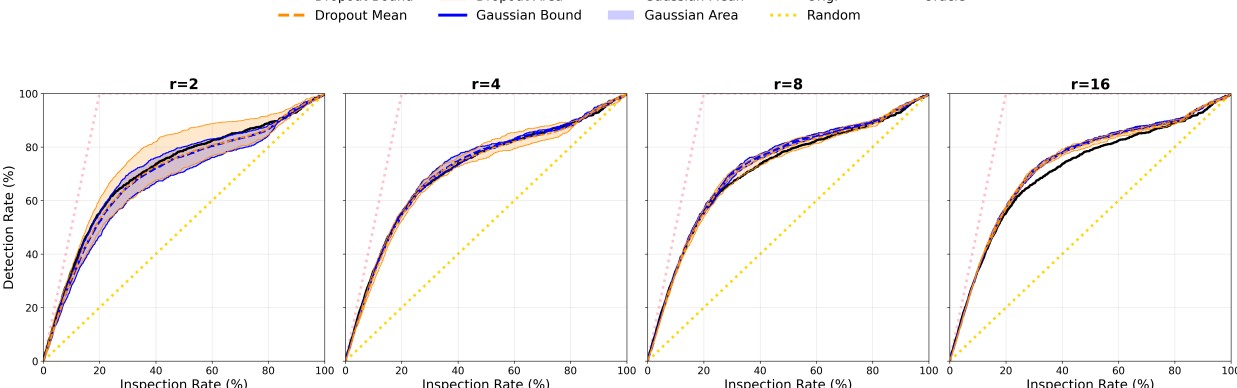

Figure 2: Mislabeled data detection on COLA (one benchmark in GLUE) with 2-rank LoRA. We compare `Orig` with gradient compression methods `Gaussian` and `Dropout` with different compression size $r$ (2, 4, 8, and 16). For `Gaussian` and `Dropout`, the shaded regions represent the range (min/max) of performance, and the solid lines indicate the average detection performance across 5 independent runs.

random seeds. Intuitively, `Dropout` may cause significant and random information loss, potentially leading to larger and unstable error, especially when the compression size $r$ is small. Therefore, it is valuable to investigate how influence function performance with `Dropout` varies with different compression size $r$. For this, we use mislabeled data detection as a case study.

In Figure 2, the bounds represent the best and the worst performance observed across runs. Empirical results indicate that while `Dropout` exhibits higher variance than `Gaussian` at extreme compression levels (e.g., $r = 2$), this instability diminishes rapidly as the compression size $r$ increases. In particular, for $r = 8$ and $r = 16$, our method exhibits significantly reduced variance, matching the stability of the `Gaussian` baseline while maintaining a comparable average detection rate. This demonstrates that once a minimal threshold of the parameter space is sampled, `Dropout` effectively captures the necessary signal for reliable influence estimation. Furthermore, while the upper bound provided in Section 3.3 may be loose, it still provides a meaningful indicator of robust performance.

# 6  Related Works

Data attribution aims to quantify and understand the impact of each training data point on the performance of the model (Albalak et al., 2024). In Ghorbani & Zou (2019), the authors proposed Data Shapley, which quantifies the value of each training data point by leveraging shapley value as a metric. Despite its conceptual appeal, Data Shapley is computationally prohibitive, particularly for modern large-scale ML models (Jia et al., 2019). Furthermore, several works proposed frameworks to do data attribution by retraining a model multiple time to evaluate the impact of some data point (Ilyas et al., 2022; Park et al., 2023). Although efforts such as (Park et al., 2023) strive to balance computational cost and effectiveness, the necessity of retraining models remains a significant drawback, especially for resource intensive deep learning applications.

Influence function is another approach to data attribution, adapted from robust statistics (Law, 1986; Hampel, 1974), and introduced to the deep learning context in Koh & Liang (2017); Koh et al. (2019). It addresses the counterfactual question: how would the model's parameters or loss change if a specific training data point were removed? While influence function offers a theoretically grounded framework for data attribution, the high computational cost has limited its applicability to large-scale models. To mitigate the computational burden of influence function, various methods have been proposed. In Koh & Liang (2017); Agarwal et al. (2017), the authors introduced LiSSA, which approaches iHVP computation iteratively, reducing the cost of influence function computation. Other approaches include LOGRA (Choe et al., 2024) and EK-FAC (Grosse et al., 2023) proposed using Kronecker product for gradients computation, and compress gradients using Kronecker product structure or eigen decompositions for efficiency. However, the Kronecker product structure cannot be universally applied to all deep learning models and eigen decompositions will be expensive in large-scale matrix. DataInf (Kwon et al., 2023) proposed an approximation of the influence function by approximating the inverse Hessian matrix. However, this method introduces errors which scale quadratically with the size of the model. Consequently, DataInf is primarily optimized for low-rank adaptation (LoRA) (Hu et al., 2022) and becomes computationally intractable for full-parameter attribution in large-scale models. Zhou et al. (2024) proposed a method which approximates the Hessian matrix using Generated Fisher Information Matrix (GFIM). This approach relies on a strong assumption that each column of the gradient matrix is independent and has a zero mean, which often fails to hold in practice.

## 7 Conclusion

In this work, we demonstrate that the influence of training data on a small subset of parameters can effectively reflect its influence on the entire parameter space. Building on this, we introduce **dropout** as a compression tool to enable efficient influence function computation, addressing the computational and memory challenges that hinder the application of influence function in large-scale models. Our approach leverages this simplicity and scalability of dropout to selectively retain gradient information, thereby significantly reducing computational and memory overhead compared to methods relying on dense compression map such as gaussian compression.

Through theoretical analysis, we demonstrated that the error upper bound of influence function with dropout compression is smaller than gaussian compression methods. Our empirical results validate these findings, showing that dropout compression method could achieve comparable or superior performance in data attribution while maintaining high efficiency. This work highlights the potential of dropout as a lightweight, efficient, and practical compression tool in influence function computation, paving the way to extending application of influence function in large-scale artificial intelligence systems.

## 8 Limitations

While we demonstrate the potential of dropout as an efficient gradient compression method for influence function computation, several limitations remain to be addressed. Our method does not alleviate the resource requirements for gradient computation. Computing gradients for all data points, particularly in large-scale models and datasets, remains a bottleneck. This limitation highlights the need for further optimizations to make influence function methods more resource-efficient.

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

# A   Appendix

## A.1   Efficient Methods for Influence Functions.

**LiSSA** Agarwal et al. (2017) proposed an iterative method for computing iHVP $(\boldsymbol{H} + \lambda \boldsymbol{I}_d)^{-1} \boldsymbol{v}$, which was later utilized by (Koh & Liang, 2017) to compute the influence function. For $\boldsymbol{s}_0 = vv$, LiSSA recursively computes the following equation: $\boldsymbol{s}_{i+1} = \boldsymbol{v} + (\boldsymbol{I}_d - (\boldsymbol{H} + \lambda \boldsymbol{I}_d)) \boldsymbol{s}_i$. Agarwal et al. (2017) proved that if $\boldsymbol{H} + \lambda \boldsymbol{I}_d \preceq \boldsymbol{I}_d$, $\boldsymbol{s}_i$ will converge to the iHVP $(\boldsymbol{H} + \lambda \boldsymbol{I}_d)^{-1} \boldsymbol{v}$, as $i$ increases. Then, the iHVP could be approximated as:

$$\boldsymbol{s}_i \approx (\boldsymbol{H} + \lambda \boldsymbol{I}_d)^{-1} \boldsymbol{v}, \tag{6}$$

and the influence function could be calculated using this approximation iHVP:

$$\mathcal{I}(z_{\text{tr}}^k) = -\boldsymbol{s}_i^\top \boldsymbol{g}_{\text{tr}}^k. \tag{7}$$

Here $\boldsymbol{g}_{\text{tr}}^k = \nabla_\theta l(y_{\text{tr}}^k, f_\theta(x_{\text{tr}}^k))\big|_{\theta=\theta^*}$ is the gradient of the loss function calculated at the $k$-th training data point with respect to parameters of the model, and $\boldsymbol{v} = \frac{1}{m}\sum_{j=1}^m \boldsymbol{g}_{\text{val}}^j = \frac{1}{m}\sum_{j=1}^m \nabla_\theta l(y_{\text{val}}^j, f_\theta(x_{\text{val}}^j))\big|_{\theta=\theta^*}$ is the average of gradients of the loss function calculated at evaluation dataset with respect to parameters of the model.

**DataInf** Kwon et al. (2023) proposed an approximated version of the influence function. The key approximation in DataInf involves swapping the order of matrix inversion and the averaging in $(\boldsymbol{H} + \lambda \boldsymbol{I}_d)^{-1}$. Using this approximation, the inverse Hessian matrix becomes:

$$(\frac{1}{n}\sum_k \boldsymbol{g}_{\text{tr}}^k \boldsymbol{g}_{\text{tr}}^{kT} + \lambda \boldsymbol{I}_d)^{-1} \approx \frac{1}{n}\sum_k (\boldsymbol{g}_{\text{tr}}^k \boldsymbol{g}_{\text{tr}}^{kT} + \lambda \boldsymbol{I}_d)^{-1} \tag{8}$$

$$= \frac{1}{n\lambda}\sum_k (\boldsymbol{I}_d - \frac{\boldsymbol{g}_{\text{tr}}^k \boldsymbol{g}_{\text{tr}}^{kT}}{\lambda + \boldsymbol{g}_{\text{tr}}^{kT} \boldsymbol{g}_{\text{tr}}^k}), \tag{9}$$

where $\boldsymbol{g}_{\text{tr}}^k$ is the gradient of the loss function calculated at the $k$-th training data point with respect to parameters of the model. The Sherman-Morrison formula (Sherman, 1949) is utilized to compute the matrix inversion in equation 9. Based on this approximation, the influence function can be computed efficiently, reducing the operation to linear complexity.

**LOGRA** Grosse et al. (2023); Choe et al. (2024) proposed using Kronecker product to approximate gradients and (Choe et al., 2024) compresses gradients make use of Kronecker product structure. For the $q$-th layer of a deep learning model with parameter $\theta_q$, let $h_q$ represent the output and $g_q$ represent the pre-activated output of the $q$-th layer. The gradient of loss function evaluated on $z = (x, y)$ with respect to $\theta_q$ is given as the following:

$$\nabla_{\theta_q} l(y, f_\theta(x)) = h_{q-1} \otimes \nabla_{g_q} l(y, f_\theta(x)), \tag{10}$$

where $\otimes$ represents the Kronecker product. LOGRA (Choe et al., 2024) imposes an additional Kronecker product structure on the projection matrix $\boldsymbol{P}$ as follows:

$$\boldsymbol{P} \nabla_{\theta_q} l(y, f_\theta(x)) = (\boldsymbol{P}_{\text{in}} \otimes \boldsymbol{P}_{\text{out}})(h_{q-1} \otimes \nabla_{g_q} l(y, f_\theta(x))) \tag{11}$$

$$= \boldsymbol{P}_{\text{in}} h_{q-1} \otimes \boldsymbol{P}_{\text{out}} \nabla_{g_q} l(y, f_\theta(x)), \tag{12}$$

where $\boldsymbol{P}_{\text{in}} \in \mathbb{R}^{r_{\text{in}} \times d_{\text{in}}}$, $\boldsymbol{P}_{\text{out}} \in \mathbb{R}^{r_{\text{out}} \times d_{\text{out}}}$. In equation 12, LOGRA first projects forward and backward activations onto low-dimensional space using $\boldsymbol{P}_{\text{in}}$ and $\boldsymbol{P}_{\text{out}}$ respectively, and then reconstructs projected gradient directly from these projected activations. It is important to note that $d_{\text{in}} = d_{\text{out}} = \sqrt{d}$ and $r_{\text{in}} = r_{\text{out}} = \sqrt{r}$, making it be easy to use relatively large compression size $r$.

## A.2   Algorithm Details

In this section, we provide the detailed implementation pipeline for efficient influence function estimation using `Dropout` compression. Algorithm 1 explicitly outlines the procedure, demonstrating how the `Dropout`

compression method defined in Section 3.1 is realized in practice through efficient indexing. The procedure consists of three main phases: **i) Mask Generation:** We first sample a set of indices $\mathcal{M}$ of size $r$. This set represents the active parameters that are retained. Unlike dense projection methods (e.g., Gaussian) that require storing a projection matrix of size $r \times d$, our method only requires storing the list of $r$ integers. **ii) Gradient Compression:** For both validation and training data, we compute the full gradient but immediately discard all entries not in $\mathcal{M}$. This ensures that the storage requirement for each data point is strictly $O(r)$. **iii) Influence Estimation:** The inverse Hessian vector product (iHVP) is computed entirely within the low-dimensional subspace $\mathbb{R}^{r \times r}$. The inversion of the compressed Hessian has a complexity of $O(r^3)$, which is negligible for small $r$. By directly operating on the indices, we avoid the overhead of sparse matrix multiplication, achieving true $O(r)$ memory and computational efficiency.

---

**Algorithm 1** Efficient Influence Function Estimation via `Dropout` Compression

---

**Input:** Training dataset $\mathcal{D}_{tr}$, Validation dataset $\mathcal{D}_{val}$, Model parameters $\theta^*$, Compression size $r$, Damping $\lambda$.

**Output:** Approximate Influence scores $\{\tilde{\mathcal{I}}(z_{tr}^k)\}$.

1: **// Step 1: Generate Compression Mask**
2: Randomly sample a set of indices $\mathcal{M} \subset \{1, \ldots, d\}$ such that $|\mathcal{M}| = r$.
3: *This corresponds to the non-zero diagonal entries of $\tilde{I}^\top \tilde{I}$.*
4: **// Step 2: Compress Validation Gradient**
5: Compute full validation gradient: $g_{val} \leftarrow \frac{1}{m} \sum_{(x_{\text{val}}, y_{\text{val}}) \in \mathcal{D}_{val}} \nabla_\theta \ell(y_{\text{val}}, f_\theta(x_{\text{val}}))|_{\theta=\theta^*}$.
6: Store only sampled indices: $\tilde{g}_{val} \leftarrow g_{val}[\mathcal{M}]$    {Memory cost: $O(r)$}
7: **// Step 3: Compute Compressed Hessian Inverse**
8: Compute Hessian sub-matrix on indices $\mathcal{M}$: $\tilde{\boldsymbol{H}}$.
9: Compute inverse term: $\boldsymbol{H}_{inv} \leftarrow (\tilde{\boldsymbol{H}} + \lambda \boldsymbol{I}_r)^{-1}$    {Compute cost: $O(r^3)$}
10: **// Step 4: Compute Influence for Training Data**
11: **for** each $z_{tr}^k \in \mathcal{D}_{tr}$ **do**
12:     Compute gradient $g_{tr}^k$.
13:     Compress immediately: $\tilde{g}_{tr}^k \leftarrow g_{tr}^k[\mathcal{M}]$    {Memory cost: $O(r)$}
14:     Compute score: $\tilde{\mathcal{I}}(z_{tr}^k) \leftarrow -\tilde{g}_{val}^\top \boldsymbol{H}_{inv} \tilde{g}_{tr}^k$    {Compute cost: $O(r^2)$}
15: **end for**
16: **return** $\{\tilde{\mathcal{I}}(z_{tr}^k)\}_k$

---

## A.3 Complexity Comparison.

Table 7 presents a comparison of the computational and memory complexity of iHVP computation for influence estimation across `Orig`, `LiSSA`, `DataInf` and gradient compression methods with compressed size $r$, such as `Gaussian`, `LOGRA`, `FJLT` and `Dropout`. While gradient compression methods cannot reduce the complexity to linear, the compression size $r$ could be very small, making these methods efficient in practice.

Although the computational and memory complexity involved in iHVP computation are the same for gradient compression methods with the same compressed size $r$, the complexity of compression process itself differ. Table 8 provides a comparison across `Gaussian`, `LOGRA`, `FJLT` and `Dropout`.

## A.4 Proof of Theorems.

In this section, we provide the formal statements and detailed proofs for the error bounds discussed in Section 3.3. We begin by introducing the fundamental lemmas and identities required for our analysis.

Table 7: Comparison of computational and memory complexity involved in iHVP computation for influence function estimation across `Orig`, `LiSSA`, `DataInf` and gradient compression methods with compression size $r$ (e.g. `Gaussian`, `LOGRA`, `FJLT` and `Dropout`). The number of parameters in the model is $d$ and the number of data is $n$.

| Method | Computational Complexity | Memory Complexity |
|---|---|---|
| Orig. | $O(nd^2 + d^3)$ | $O(d^2)$ |
| LiSSA | $O(nd^2)$ | $O(d^2)$ |
| DataInf | $O(nd)$ | $O(d)$ |
| Compressing Methods | $O(nr^2 + r^3)$ | $O(r^2)$ |

Table 8: Comparison of computational and memory complexity involved in performing compression across `Gaussian`, `LOGRA`, `FJLT` and `Dropout`. The number of parameters in the model is $d$, the number of data is $n$, and the compressed size is $r$.

| Method | Computational Complexity | Memory Complexity |
|---|---|---|
| Gaussian | $O(nrd)$ | $O(rd)$ |
| LOGRA | $O(n\sqrt{rd})$ | $O(\sqrt{rd})$ |
| FJLT | $O(d\log d)$ | $O(d)$ |
| Dropout | $O(nr)$ | $O(r)$ |

---

**Theorem A.1: Woodbury Matrix Identity (Harville, 1998)**

Given a square invertible $n \times n$ matrix $\boldsymbol{A}$, an $n \times k$ matrix $\boldsymbol{U}$, and a $k \times n$ matrix $\boldsymbol{V}$, let $\boldsymbol{B} = \boldsymbol{A} + \boldsymbol{U}\boldsymbol{V}$. If $(\boldsymbol{I}_k + \boldsymbol{V}\boldsymbol{A}^{-1}\boldsymbol{U})$ is invertible, then:

$$\boldsymbol{B}^{-1} = \boldsymbol{A}^{-1} - \boldsymbol{A}^{-1}\boldsymbol{U}(\boldsymbol{I}_k + \boldsymbol{V}\boldsymbol{A}^{-1}\boldsymbol{U})^{-1}\boldsymbol{V}\boldsymbol{A}^{-1}. \tag{13}$$

---

The theorem of woodbury matrix identity A.1 not only allows cheaper computation of inverses but also provides a closed form expression of matrix inversion.

---

**Theorem A.2: Convergence of Extreme Singular Values (Bai et al., 1993)**

Let $\boldsymbol{A}$ be a $k \times n$ random matrix whose entries are independent copies of a random variable with zero mean, unit variance, and finite fourth moment. If $k, n \to \infty$ with $n/k \to \kappa \in (0, 1]$, then:

$$\frac{1}{\sqrt{k}}\sigma_{\max}(\boldsymbol{A}) \to 1 + \sqrt{\kappa} \quad \text{almost surely.} \tag{14}$$

---

Theorem A.2 implies that for a $k \times n$ random Gaussian matrix, the largest singular value $\sigma_{\max}$ converges to $O(\sqrt{k})$.

### A.4.1 Theorem for Gaussian Compression

---

**Theorem A.3: Gaussian Compression Error Bound**

For gaussian based compression method in equation 2, if $\lambda \boldsymbol{I}_d + \boldsymbol{P}^\top \boldsymbol{P} \boldsymbol{H}$ is invertible and the dimension of $\theta$ exceeds the size of training dataset $n$, the spectral norm of $\Delta \boldsymbol{H}$, i.e. $\left\|\Delta \boldsymbol{H}\right\|_2$, is bounded by:

$$O(d + d^2 \sigma_{\max}(\boldsymbol{H})),$$

where $\sigma_{\max}(\boldsymbol{H})$ denotes the largest singular value of $\boldsymbol{H}$.

---

*Proof.* Using the Woodbury identity in equation 13, we express the error matrix $\Delta \boldsymbol{H}$ as:

$$\Delta \boldsymbol{H} = (\lambda \boldsymbol{I}_d + \boldsymbol{H})^{-1} - \boldsymbol{P}^\top (\lambda \boldsymbol{I}_r + \boldsymbol{P} \boldsymbol{H} \boldsymbol{P}^\top)^{-1} \boldsymbol{P}$$

$$= (\lambda \boldsymbol{I}_d + \boldsymbol{H})^{-1} - \frac{1}{\lambda} \boldsymbol{P}^\top \boldsymbol{P} + \frac{1}{\lambda} \boldsymbol{P}^\top \boldsymbol{P} \boldsymbol{H} (\lambda \boldsymbol{I}_d + \boldsymbol{P}^\top \boldsymbol{P} \boldsymbol{H})^{-1} \boldsymbol{P}^\top \boldsymbol{P}. \tag{15}$$

Applying the triangle inequality and properties of the spectral norm:

$$\|\Delta \boldsymbol{H}\|_2 \leq \|(\lambda \boldsymbol{I}_d + \boldsymbol{H})^{-1}\|_2 + \frac{1}{\lambda} \|\boldsymbol{P}^\top \boldsymbol{P}\|_2 + \frac{1}{\lambda} \|\boldsymbol{P}^\top \boldsymbol{P} \boldsymbol{H}\|_2 \|(\lambda \boldsymbol{I}_d + \boldsymbol{P}^\top \boldsymbol{P} \boldsymbol{H})^{-1}\|_2 \|\boldsymbol{P}^\top \boldsymbol{P}\|_2$$

$$\leq \frac{1}{\sigma_{\min}(\lambda \boldsymbol{I}_d + \boldsymbol{H})} + \frac{\sigma_{\max}(\boldsymbol{P}^\top \boldsymbol{P})}{\lambda} + \frac{\sigma_{\max}(\boldsymbol{P}^\top \boldsymbol{P})^2 \sigma_{\max}(\boldsymbol{H})}{\lambda \sigma_{\min}(\lambda \boldsymbol{I}_d + \boldsymbol{P}^\top \boldsymbol{P} \boldsymbol{H})}. \tag{16}$$

Since the model parameters $d$ exceed the dataset size $n$, $\boldsymbol{H}$ is not full rank, so $\sigma_{\min}(\lambda \boldsymbol{I}_d + \boldsymbol{H}) = \lambda$. From Theorem A.2, we have $\sigma_{\max}(\boldsymbol{P}^\top \boldsymbol{P}) \leq \sigma_{\max}(\boldsymbol{P})^2 \leq d$. Substituting these into equation 16:

$$\|\Delta \boldsymbol{H}\|_2 \leq \frac{1}{\lambda} + \frac{d}{\lambda} + \frac{d^2 \sigma_{\max}(\boldsymbol{H})}{\lambda^2} \propto O(d + d^2 \sigma_{\max}(\boldsymbol{H})). \tag{17}$$

$\square$

### A.4.2 Theorem for Dropout Compression

---

**Theorem A.4: Dropout Compression Error Bound**

For dropout based compression method in equation 4, if the dimension of $\theta$ exceeds the size of training dataset $n$, the spectral norm of $\Delta \boldsymbol{H}$, i.e. $\left\|\Delta \boldsymbol{H}\right\|_2$, is bounded by:

$$O(\sigma_{\max}(\boldsymbol{H})),$$

where $\sigma_{\max}(\boldsymbol{H})$ denotes the largest singular value of $\boldsymbol{H}$.

---

*Proof.* Let $\tilde{\boldsymbol{I}}$ be the binary selection matrix defined in equation 3. Using the Woodbury identity:

$$\Delta \boldsymbol{H} = (\lambda \boldsymbol{I}_d + \boldsymbol{H})^{-1} - \frac{1}{\lambda} \tilde{\boldsymbol{I}}^\top \tilde{\boldsymbol{I}} + \frac{1}{\lambda} \tilde{\boldsymbol{I}}^\top \tilde{\boldsymbol{I}} \boldsymbol{H} (\lambda \boldsymbol{I}_d + \tilde{\boldsymbol{I}}^\top \tilde{\boldsymbol{I}} \boldsymbol{H})^{-1} \tilde{\boldsymbol{I}}^\top \tilde{\boldsymbol{I}}. \tag{18}$$

Taking the spectral norm:

$$\|\Delta \boldsymbol{H}\|_2 \leq \frac{1}{\sigma_{\min}(\lambda \boldsymbol{I}_d + \boldsymbol{H})} + \frac{\sigma_{\max}(\tilde{\boldsymbol{I}}^\top \tilde{\boldsymbol{I}})}{\lambda} + \frac{\sigma_{\max}(\tilde{\boldsymbol{I}}^\top \tilde{\boldsymbol{I}})^2 \sigma_{\max}(\boldsymbol{H})}{\lambda \sigma_{\min}(\lambda \boldsymbol{I}_d + \tilde{\boldsymbol{I}}^\top \tilde{\boldsymbol{I}} \boldsymbol{H})}$$

$$\leq \frac{1}{\lambda} + \frac{1}{\lambda} + \frac{\sigma_{\max}(\boldsymbol{H})}{\lambda^2}. \tag{19}$$

Table 9: Hyperparameters used for model training in experiments.

|  | RoBERTa | ResNet-9 | GPT-2 | Pythia 1.4B | Pythia 6.9B |
|---|---|---|---|---|---|
| Optimizer | AdamW | SGD-M | AdamW | AdamW | AdamW |
| LR Scheduler | Linear | Linear | None | None | Cosine |
| Learning Rate | $3 \times 10^{-4}$ | 0.4 | $5 \times 10^{-5}$ | $2 \times 10^{-1}$ | $5 \times 10^{-5}$ |
| Weight Decay | None | $5 \times 10^{-4}$ | 0.01 | 0.01 | 0.01 |
| Batch Size | 32 | 64 | 64 | 16 | 2 |
| Sequence Length | None | None | 256 | 128 | 1024 |
| Epochs | 10 | 24 | 3 | 1 | 1 |

Table 10: Average time usage (seconds) of computing iHVP in mislabeled data detection tasks.

| Method | Rank=2 | | | | Rank=8 | | | |
|---|---|---|---|---|---|---|---|---|
|  | MRPC | QNLI | QQP | SST2 | MRPC | QNLI | QQP | SST2 |
| Orig | 319.22 | 384.26 | 335.40 | 418.29 | - | - | - | - |
| LiSSA | 43.05 | 53.65 | 53.77 | 53.46 | 74.19 | 83.76 | 84.54 | 84.68 |
| DataInf | 7.97 | 9.85 | 10.16 | 9.89 | 11.17 | 13.38 | 13.54 | 13.45 |
| Gaussian | 3.74 | 4.58 | 4.62 | 4.60 | 3.83 | 4.72 | 4.75 | 4.71 |
| PCA | 3.80 | 4.59 | 4.67 | 4.87 | 3.86 | 4.77 | 4.82 | 4.82 |
| Dropout(Ours) | 3.84 | 4.68 | 4.75 | 4.73 | 3.93 | 4.83 | 4.87 | 4.85 |

Because $\tilde{\boldsymbol{I}}$ is a subset selection matrix, its singular values are exactly 1 or 0, thus $\sigma_{\max}(\tilde{\boldsymbol{I}}^{\top}\tilde{\boldsymbol{I}}) = 1$. This results in:

$$\|\Delta \boldsymbol{H}\|_2 \propto O(\sigma_{\max}(\boldsymbol{H})). \tag{20}$$

This bound is independent of the parameter dimension $d$, explaining the stability of dropout-based compression for large-scale models. $\square$

### A.5 Experiments.

#### A.5.1 Details of Experiments.

For each methods, we set the damping term in influence function as $\lambda_l = 0.1 \times (nd_l)^{-1} \sum_{i=1}^{n} \nabla_{\theta_l} l_i^{\top} \nabla_{\theta_l} l_i$ for layer $l$, where $\theta_l$ represents the parameters of the $l$-th layer, $\nabla_{\theta_l} l_i$ represents the gradient of the loss function calculated at the $i$-th data point with respect to $\theta_l$, and $d_l$ represents the number of parameters in this layer. For model training, we use hyperparameters in Table 9.

#### A.5.2 More Results.

In this section, we include more results of experiments. Table 10 contains the average time usage for computing iHVP in mislabeled data detection tasks. Table 11 presents the average time usage for gradients compression in retraining ResNet-9 and GPT-2. ResNet-9 has fewer than 1 million parameters, whereas GPT-2-small exceeds 100 million parameters. As a result, applying `PCA` or `Gaussian` becomes computationally expensive for GPT-2. To address this, we use the more efficient `FJLT` method for comparison. The results demonstrate that `Dropout` is highly efficient during the compression process while preserving the effectiveness of the influence function.

#### A.5.3 Examples of Identified Influential Data.

To qualitatively assess whether our method identifies meaningful influential examples, we present representative cases where the most influential training samples retrieved by our method align with the content

Table 11: Average time usage (seconds) of compression gradients in retraining ResNet-9 and GPT-2.

| Model | PCA | Gaussian | FJLT | Dropout |
|---|---|---|---|---|
| **ResNet-9** | 1344.34 | 235.64 | - | 8.44 |
| **GPT-2** | - | - | 453.87 | 9.76 |

or source of a given validation example. We list some examples in the following. These examples were selected from experiments using the Pythia-6.9B model fine-tuned on a heterogeneous dataset composed of six sources.

---

**Example A.1**

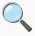 **Test Example**
Absolute quiet descended on the square. Fourteen mech wolves lay out of commission and more than thirty freedom fighters. Mech bird corpses were scattered everywhere. On the platform, Ajax curled his upraised hands into fists. "I, Ajax, warrior hero, hereby assume my rightful place as leader of Mech City!" No one dared disagree. "I order all wolves to stand down! All two-legged robots cease fighting – immediately!" The freedom fighters lowered their weapons. The birds took to the sky en masse. The wolves glanced sheepishly at each other, unwilling to disobey Ajax's command despite their previous orders. Looking out from his place of concealment, Fascista Ultimo viewed their hesitation with alarm. He uttered a directive into his transmitter, and the mech wolves retreated to the square's periphery.

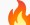 **Most Influential Training Example**
This is my home you so arrogantly took up residence in." Michael nodded at Liz. "This is my mate that the lot of you tried to kill." With another nod to Avery, he finished. "And last but not least, this is my charge–that you beat and tried to feed upon. What do you propose we do about that, Carl?" Words gushed from the tattered young vampire's mouth. "We thought you were dead. All of us did. We were told that you died in the fortress!" "Who told you I was dead?" "The Council. They said that you, and all the rogues with you, died when the fortress collapsed." Michael laughed at his answer, and Liz laughed with him. "So Monroe is calling us the rogues now, is he? months old, she would bet. She felt no malice from them, just fright and insecurity.

---

**Example A.2**

🔍 **Test Example**

Carl Froch's hopes of fighting Julio Cesar Chavez Jnr in a 50 million Las Vegas send-off have suffered a potential setback. The Mexican's promoter, Bob Arum, is currently negotiating a new deal with his fighter and expects him to return in September or October. But he wants Chavez to face the hard-hitting Kazakh Gennady Golovkin, rather than Froch, as reported by Boxing Scene. Out cold: Carl Froch stopped George Groves in the eighth round of their rematch at Wembley . 'Gennady Golovkin will continue to be the first choice assuming he beats [Daniel] Geale, and if Chavez says no to Golovkin, then we'll look to Froch,' Arum told Boxing Scene. Froch retained his super-middleweight world titles last weekend with a stunning eighth-round knockout of George Groves in front of 80,000 fans at Wembley. The Nottingham Cobra has admitted he could hang up his gloves but dreams of finishing his career in the fight capital of the world. But he might have to wait until next year if Chavez agrees to fight Golovkin, a fight that had been mooted for this summer before negotiations stalled. Option: Froch has targetted a Las Vegas send-off against Julio Cesar Chavez Jnr. Golovkin instead defends his WBA Super and IBO middleweight titles against former world champion Geale in New York's Madison Square Garden on July 26. And Golovkin's trainer Abel Sanchez is confident his charge would beat Froch if the two were to meet. 'Froch is a gallant warrior, but makes too mistakes and if the fight can be made, I see Golovkin capitalising on them to stop him in the last part of the fight,' he told World Boxing News. Another option for Froch is another domestic dust-up with mandatory challenger James DeGale. The IBF has ruled that Froch must face the Londoner within the nine months or be stripped of the belt. Domestic: James DeGale is Froch's mandatory challenger after beating Brandon Gonzalez last weekend .

🔥 **Most Influential Training Example**

(CNN) – Some of the president's men are still working. In golf's Presidents Cup, that is. And while U.S. President Barack Obama wasn't in attendance in Ohio – he has more important things to worry about – a former president, George W. Bush, greeted both teams Thursday at the Muirfield Village Golf Club. The biennial competition, which pits the U.S. against the rest of the world minus Europe, has been kinder to the Americans than the more prestigious Ryder Cup. Indeed since the tournament started in 1994, only once has the International Team prevailed, and the U.S. has won four in a row. The U.S. featured six players in the top 10 in the rankings, including world No. 1 and PGA Player of the Year Tiger Woods. The International Team, by contrast, had one – Masters champion Adam Scott. Early indications suggested the U.S.'s superiority in the rankings would translate to an easy victory – they led all six fourballs in the early stages. But after about a 90-minute delay because of thunderstorms, the International Team fought back. By day's end, it was 3.5 to 2.5 for the U.S., a slender advantage. Jason Day's dramatic putt at the 18th gave the Australian and Graham DeLaet a win over Hunter Mahan and Brandt Snedeker after Woods and Matt Kuchar routed Angel Cabrera and Marc Leishman 5 and 4. "It's awfully fun partnering the No. 1 player in the world," Kuchar said in a televised interview. The U.S. fell behind as Phil Mickelson and Keegan Bradley lost to Louis Oosthuizen and Charl Schwartzel 2 and 1 before Scott and Hideki Matsuyama halved with Bill Haas and Webb Simpson. Jason Dufner and Zach Johnson eased past Branden Grace and Richard Sterne 5 and 3 to level the match. In the decider, Steve Stricker and Jordan Spieth edged Ernie Els and Brendon de Jonge. Stricker's short putt at the 18th gave the U.S. the lead heading into Friday's foursomes.

---

**Example A.3**

🔍**Test Example**

1. Let $n$ be the total number of students, which is given as 33. 2. Let $A$ be the initial average mark of the class. 3. The total sum of marks for all students is $n \times A = 33 \times A$. 4. The total sum of marks for the 3 students with an average mark of 40 is $3 \times 40 = 120$. 5. After excluding these 3 students, the remaining number of students is $n - 3 = 33 - 3 = 30$. 6. The new average mark for the remaining 30 students is given as 95. 7. The total sum of marks for the remaining 30 students is $30 \times 95$. 8. The total sum of marks for all 33 students is the sum of the marks of the 3 excluded students plus the sum of the marks of the remaining 30 students:

$$33 \times A = 120 + 30 \times 95$$

9. Calculate the sum:
$$30 \times 95 = 2850$$

10. Substitute back into the equation:

$$33 \times A = 120 + 2850 = 2970$$

11. Solve for $A$:
$$A = \frac{2970}{33} = 90$$

12. Therefore, the initial average mark of the class was 90.

🔥**Most Influential Training Example**

1. Let $V$ be the total number of valid votes. 2. Let $T$ be the total number of votes cast in the election. 3. Given that one candidate got 60% of the total valid votes, this means the other candidate got 40% of the total valid votes. 4. We know that the other candidate got 2519.9999999999995 valid votes, which is 40% of the total valid votes. 5. Therefore, we can set up the equation:

$$0.40 \times V = 2519.9999999999995$$

6. Solving for $V$:
$$V = \frac{2519.9999999999995}{0.40} = 6299.999999999999$$

7. Since $V$ represents the total valid votes, we can approximate $V$ to 6300 (as the fraction is extremely close to an integer). 8. We also know that 30% of the total votes were invalid, so 70% of the total votes were valid. 9. Therefore, the valid votes $V$ are 70% of the total votes $T$:

$$0.70 \times T = 6300$$

10. Solving for $T$:
$$T = \frac{6300}{0.70} = 9000$$

11. Thus, the total number of votes in the election is 9000.

---

**Example A.4**

🔍**Test Example**
1. **Identify the total number of marbles**: - There are 4 red marbles, 3 blue marbles, and 6 green marbles. - Total number of marbles = 4 (red) + 3 (blue) + 6 (green) = 13 marbles. 2. **Identify the number of favorable outcomes**: - We are interested in drawing a marble that is either red or blue. - Number of red marbles = 4. - Number of blue marbles = 3. - Total number of red or blue marbles = 4 (red) + 3 (blue) = 7 marbles. 3. **Calculate the probability**: - The probability of drawing a marble that is either red or blue is the number of favorable outcomes divided by the total number of marbles. - Probability = Number of red or blue marbles / Total number of marbles = 7 / 13. 4. **Express the probability as a fraction**: - The fraction is already in its simplest form, which is $\frac{7}{13}$. Therefore, the probability of drawing a marble which is either red or blue is $\frac{7}{13}$.

🔥**Most Influential Training Example**
1. **Identify the total number of students**: - Total students = 300 2. **Calculate the number of students who got first division**: - Percentage of first division students = 29% - Number of first division students = $29\% \times 300 = \frac{29}{100} \times 300 = 87$ students 3. **Calculate the number of students who got second division**: - Percentage of second division students = 54% - Number of second division students = $54\% \times 300 = \frac{54}{100} \times 300 = 162$ students 4. **Calculate the number of students who just passed**: - The remaining students are those who just passed. - Total students who got first and second division = 87 (first division) + 162 (second division) = 249 students - Number of students who just passed = Total students - Students with first and second division - Number of students who just passed = 300 - 249 = 51 students 5. **Conclusion**: - The number of students who just passed is 51.

---

**Example A.5**

🔍**Test Example**
The best treatment advice for a 69-year-old male with coronary artery disease who has asymptomatic gallbladder stones, discovered during a routine ultrasound and with no history of biliary colic or jaundice, is: C. No surgery for gallbladder stones In cases where gallbladder stones are asymptomatic, the standard approach is usually conservative management, monitoring the condition without immediate surgery. This strategy helps avoid unnecessary surgical risks, particularly for patients with additional health concerns such as coronary artery disease. Surgery is generally considered only if the stones become symptomatic.

🔥**Most Influential Training Example**
Given the clinical presentation and history, the most appropriate immediate treatment for this patient is intravenous hydrocortisone. Her symptoms—severe abdominal pain, confusion, vomiting, high temperature, rapid pulse, and dangerously low blood pressure unresponsive to fluids—suggest an adrenal crisis, especially in the context of an autoimmune condition like Hashimoto thyroiditis, which is associated with an increased risk of other autoimmune disorders such as Addison's disease (adrenal insufficiency). During times of stress, such as after surgery, an adrenal crisis can occur if there is inadequate cortisol production. Administering IV hydrocortisone would address the cortisol deficiency and help stabilize her condition.

---

**Example A.6**

🔍**Test Example**
table: 1-1137718-2 columns: Rd., Grand Prix, Date, Location, Pole Position, Fastest Lap, Winning Driver, Constructor, Report Q: Who was the winning driver on 13 August? A: SELECT Winning Driver FROM 1-1137718-2 WHERE Date = '13 August'

🔥**Most Influential Training Example**
table: 1-1139087-2 columns: Rd., Grand Prix, Date, Location, Pole Position, Fastest Lap, Winning Driver, Constructor, Report Q: What day was the grand prix in jerez? A: SELECT Date FROM 1-1139087-2 WHERE Location = 'Jerez'

---

**Example A.7**

🔍**Test Example**
Given the clinical scenario of a young woman experiencing significant cardiac symptoms such as syncope, bradycardia, and palpitations, especially with exertion, in the context of confirmed Lyme disease, the most appropriate treatment is intravenous antibiotics. The bradycardia and syncope suggest that Lyme carditis may have caused a high-degree atrioventricular (AV) block, which requires prompt and aggressive treatment to prevent complications. In these cases, intravenous ceftriaxone is typically recommended as it is highly effective at addressing more severe manifestations of Lyme disease, particularly those involving the heart. This treatment approach ensures that the infection is rapidly controlled, reducing the risk of further cardiac issues. Therefore, for this patient with confirmed Lyme disease and significant cardiac involvement, the most appropriate treatment is IV ceftriaxone.

🔥**Most Influential Training Example**
Given the clinical presentation and history, the most appropriate immediate treatment for this patient is intravenous hydrocortisone. Her symptoms—severe abdominal pain, confusion, vomiting, high temperature, rapid pulse, and dangerously low blood pressure unresponsive to fluids—suggest an adrenal crisis, especially in the context of an autoimmune condition like Hashimoto thyroiditis, which is associated with an increased risk of other autoimmune disorders such as Addison's disease (adrenal insufficiency). During times of stress, such as after surgery, an adrenal crisis can occur if there is inadequate cortisol production. Administering IV hydrocortisone would address the cortisol deficiency and help stabilize her condition.

---

**Example A.8**

🔍**Test Example**

By . Daily Mail Reporter . PUBLISHED: . 00:57 EST, 6 November 2012 . | . UPDATED: . 10:57 EST, 7 November 2012 . Four years on and Katie Holmes was back on the Broadway stage on Monday night. Reaction to her turn in Dead Accounts is yet to get going, but the actress certainly ensured she got back to work with a bang by sharing a steamy smooch with co-star Josh Hamilton. In one scene, the 33-year-old actress is seen locking lips with the fellow Thespian and throwing her arms around him. First night: Katie Holmes took to the stage on Monday evening for the preview opening night of her Broadway play Dead Accounts in New York . Passionate: The actress shared on steamy clinch with co-star Josh Hamilton in one scene . Predictably, perhaps, gossipy reports . have surfaced saying Hamilton - who is actually married to producer Lily . Thorne - has a 'crush' on Katie, according to the Daily News at least. However, such talk has been laughed off by other publications, with the Chicago Sun-Times quoting a source brandishing the rumours as 'just more scandal press garbage'. Katie looks like a girl next door in . the stills released from her performance; she is wearing a casual . ensemble of jeans, a purple top and floral cardigan, teamed with a pair .

🔥**Most Influential Training Example**

By . Jessica Jerreat . PUBLISHED: . 21:14 EST, 10 August 2013 . | . UPDATED: . 22:39 EST, 10 August 2013 . After concluding one last bit of official business, the Obamas departed The Disabled American Veteran National Convention in Florida for their their annual eight-day vacation on Martha's Vineyard, where they are staying at a 7.6 million vacation home with views of the Atlantic. Getting on the plane from Orlando the Obama's sported a smart look with the President donning a suit and Michelle Obama meticulously attired with pearls and a belt around her sun dress. However, wheh the first couple disembarked in Martha's Vineyard they were ready for their vacation to start, as Obama had changed into a pair of chinos and Michelle ditched the pearls and belt. Before: President Barack Obama and first lady Michelle Obama wave goodbye as they leave Orlando for a family vacation at Martha's Vineyard . After: The President and first lady arrive in Martha's Vineyard in more casual attire, wearing a pair of chinos and no jewelry, respectively .

