# OpenReview forum: "Toward Efficient Influence Function: Dropout as a Compression Tool"
_TMLR — Accepted by TMLR_

### Review · Reviewer_ncDR · 2025-10-05

**Summary Of Contributions:**

Modern machine learning models are heavily dependent on the quality of training data. In the statistical literature, influence function measures the influence of a single data point to the optimal parameter. However, computing the influence function requires computing the inverse hessian vector product of the parameters with respect to the loss function, which is prohibitively expensive for modern machine learning models. To address the efficiency issue, existing works propose approximation methods and compression methods to reduce the computation. This work fall into the second category. It reduces the computation by randomly dropping parameters to evaluate, and works with a much smaller hessian matrix, similar in spirit as Dropout. Theories suggest that the error of their compression method is controlled.

The experiments have three parts. First, on benchmarks of mislabeled data detection, it is found that compression methods generally work better than approximation methods, and the proposed Dropout scheme works similar to random projection with similar hyperparameter settings. Then, the behavior of the models after removing the most influential data are studied under different approximated/compressed influence functions. Dropout performs comparable to existing methods on both small- and large-scale settings. Finally, as the theoretical results are not strong enough, experiments that establish the theoretical properties are presented. The proposed Dropout scheme has a bound that is better than random projection on average, but may have higher variance with a high compression rate.

### Strengths

- The motivation of the idea is strong. Machine learning models are over-parameterized, and dropout is a good tool to utilize the power of over-parameterized neural networks while being more efficient in complicated analysis such as hessian matrices. I like that the idea is motivated by interesting properties of deep neural networks but has a different consideration.
- The proposed scheme is motivated by influence function, but can generalize to other methods. In particular, the hessian of the loss function has been used in second-order optimization, Bayesian deep learning, and many other ideas. It would be interesting to see how the proposed method could influence other areas, and what are the fundamental principles of second-order methods.

### Weaknesses

- I think the theories are quite weak. The two theorems are proved without consideration of the important hyperparameter $r$. Without the term, the comparison of the two methods could not be established and the rate of efficiency is not established.
- The exact procedure of the proposed method is unclear to me. For practitioners to use this work, there should be an algorithmic block that takes inputs and hyperparemeters, and gives the outputs. Moreover, the connection of Dropout and random projection could be inferred from the texts, but is not seriously descussed.

**Audience:**

Yes

**Audience Explanation:**

People working on anomaly detection would be interested in novel methods of detecting outliers with the method. The dropout idea could also be potentially added to the toolboxes of modern statistical workflows, once the theoretical properties are more seriously established.

Also, in applied AI, data cleaning is an important step but the methods are rather unsystematic. The line of works gives an insightful guidance of how to automate data cleaning in this era, when companies are working with trillions of datapoints daily.

In Bayesian deep learning, the hessian of the model log likelihood is used for obtaining the posterior distribution of the parameters. I believe the line of work could give insights to other methods that use the hessian of the loss functions in modern machine learning models.

**Broader Impact Concerns:**

The broader impact concerns are minor.

**Claims And Evidence:**

No

**Claims Explanation:**

- I am frustrated by the the two theoretical results. Theorem 3.1 for random projection is extremely loose. Random projection wouldn't work at all if it follows this bound. Also, intuitively as $\frac{r}{d}\to 1$, the error should go to $0$, which is not established in the theories.

- Section 5 is used to compare the two bounds empirically. However, it only tested $r=1,2,3.4$, but the models were tested with $r=16$ in other experiments.

- Section 4.3 is very confusing. I don't understand why "the influence signal should align with the semantic or source-level structure of the
dataset", and what top-1 means in the context. More explanation should be provided in this part.

Other than the above points, I think the claims are supported by good evidence, and I am convinced that this is a valuable work.

**Requested Changes:**

- If it is possible, I would like to have the theorems that include the term $r$ in the bound.
- An algorithmic block of the Dropout pipeline for computing the influence functions would be helpful for practitioners to use the methods. There is a claim "and reduce memory and computational costs to O(r) as only r entries of gradients are sampled and stored", which is not clear from the presentation.
- I'd like to see more experiments in Section 5. For example, $r=1,2,4,8,16$ would be a better selection of hyperparameters to compare.
- Please explain Section 4.3 better. Also, why is Dropout quite different from random projection in this experiment? I don't have the intuition here.

Minors:
- At the end of page 2, "mode" -> "model".
- The notations of the methods are not consistent. Both Dropout and random projection use the same $\tilde{\mathcal{I}}$, which complicates the description in the texts. Would it be possible to add subscripts to the notations, or change one of them to $\hat{\mathcal{I}}$?
- There is a minor error in the proofs of the theorems. When Woodbury matrix identity is used, the sign of the longer term is minus, which should give a plus term after plugging in. This doesn't affect the results, but should be corrected.

---

> ### Author Response · Authors · 2025-12-10
>
> We sincerely thank the reviewer for the thoughtful and detailed assessment of our work, and we appreciate the reviewer’s insight regarding the potential for our method to generalize to other second-order applications. We have prepared a comprehensive plan to address these concerns.
> 1. Clarifying the nature of theoretical bounds.\
> We deeply appreciate the reviewer highlighting this critical point. We wish to clarify that the bounds presented in Theorems 3.1 and 3.2 are intended to be worst-case upper bounds that establish a fundamental stability guarantee. Specifically, because $\sigma_{max}(\tilde{I}^T\tilde{I}) = 1$ in the case of Dropout, the spectral bound remains controlled even in extreme compression scenarios (small $r$). While this worst-case bound is indeed agnostic to the compression rate (ensuring safety), we acknowledge it may be loose for larger $r$. To bridge this gap between theory and practice, we discuss the impacts of $r$ in section 5. We explicitly stated in the revised paper that Theorem 3.2 represents a universal stability guarantee in section 3.3 and marked as blue.
> 2. Enhancing clarity for practitioners.\
> We agree that an explicit algorithmic block is helpful for ease of implementation and practical use. We added a dedicated algorithm block clearly detailing the pipeline for computing influence function with dropout in appendix A.2 and mentioned this in section 3.1. This will also clarify the claim regarding the memory and computational reduction.
> 3. Strengthening empirical comparison.\
> We acknowledge that a more comprehensive hyperparameter study in section 5 would strengthen the empirical evidence. This section is intended to show that as r increases, the fluctuation range becomes smaller. We conducted additional experiments with r=2, 4, 8, 16 and the results are attached. When r=8 and r=16, both methods become quite stable, with the dropout-based method consistently outperforming the gaussian based one. We included the new results in the revised version in section 5 and marked text as blue, and this will provide a more detailed and convincing comparison of the practical error scaling.
> 4. Improving explanation and intuition.\
> We thank the reviewer for highlighting the ambiguity in Section 4.3. We agree that clearer intuition is needed to justify this experiment.\
> Intuition: For a large model like Pythia trained on heterogeneous sources (CNN, Math Reasoning, WikiSQL, etc.), a successful influence function should ideally trace a test sample from a specific source (e.g., "Math Reasoning") back to influential training samples from that same source.\
> Metrics: The high source alignment (Table 6) demonstrates that the information preserved by Dropout is sufficient to capture these semantic structures. Top-1 measures the proportion of test cases where the single most influential training example belongs to the same source as the test example and Top-3 means the proportion of test cases where all top three most influential training examples belong to the same source. We extensively revised section 4.3 to include this detailed intuition and the definition of metrics.\
> Baseline: Regarding the "Random" baseline. We used this to establish a lower bound for performance. We explicitly excluded "Random Projection" baselines in this specific billion-scale experiment because the projection matrix $P$ would incur $O(rd)$ memory/compute costs, which is impractical for $d \approx 7B$.
> 5. Minor errors and notations.\
> We thank the reviewers for their attention to details. We corrected the sign error in the proofs of the theorem and revised the notation. We also corrected the typos.

---

> > ### Comment · Reviewer_ncDR · 2025-12-15
> >
> > Thank you for the reply. There are a few things regarding your revision:
> > - It would be misleading to call the baseline in Table 6 "Random" if it is not random projection like in the other experiments. Please come up with a better name for it.
> > - Although the bounds are still loose, the analysis in Section 5 complements this parts. However, in Figure 2, Orig works worse than Gaussian/Dropout for $r=8,16$. Do you have an explanation for this finding? Also, from this new experiments, I wouldn't say Dropout "consistently maintaining a higher average detection rate". Instead, Dropout works similar to Gaussian. It will not hurt to change the texts to match the figures.

---

> > > ### Author Response · Authors · 2025-12-16
> > >
> > > We thank the reviewer for pointing these out.
> > > 1. We agree that the term 'Random' could be confused with random projection methods. We have renamed this baseline to 'Random Ranking' throughout the paper and in Table 6. This name more accurately reflects the method assigns random influence scores to training examples and distinguishing it from 'Gaussian' baseline which utilizes random gaussian matrix projections for gradient compression.
> > > 2. We thank the reviewer for the observation that Orig works worse. We believe the superior performance of compressed methods over the original uncompressed gradients stem from a denoising effect. Full high-dimensional gradients often contain redundant or noisy information that can degrade the retrieval of truly influential examples. But compression methods act as a form of dimensionality reduction that filters our this noise. We discussed this in the result of section 4.1 Mislabeled Data Detection.
> > > 3. We have updated the text to reflect that Dropout achieves performance comparable to the Gaussian baseline rather than claiming a consistently higher detection rate.

---

### Review · Reviewer_79da · 2025-10-27

**Summary Of Contributions:**

This paper studies computationally efficient ways to compute the influence function, a concept taken from robust statistics and got modernised in recent years. Compared to previous methods which uses PCA or random Gaussian matrix, the authors proposed to use dropout as their compression mechanism to speed up the computation of the Hessian, which is the main computational bottleneck when computing the influences. Some preliminary analysis in error was provided, and thorough experiments have shown that the utility of their recovered influence scores is often on par, if not better than, other methods, while enjoying further computational efficiency.

**Audience:**

Yes

**Audience Explanation:**

Yes, I think research on the influence function is an important direction as it help people understand how data affects the model, and computational methods making such method easier and faster to run in large scale is valuable.

**Broader Impact Concerns:**

There is no ethical implications of this work.

**Claims And Evidence:**

Yes

**Claims Explanation:**

The various experiments (e.g. mislabeled data detection, model retraining) have demonstrated the utility of their method, and their reported clock time in computation has demonstrated the efficiency of their method compared to the other methods.

**Requested Changes:**

It is quite interesting to see that in Theorem 3.2, there is no mention of the number of entries being "dropped out" in the bound. I find that odd and would request an explanation in this regard. Having this will strengthen the work in my view.

---

> ### Author Response · Authors · 2025-12-10
>
> We are grateful for your positive assessment of our work. We recognize the reviewer's specific request regarding the theoretical analysis in Theorem 3.2 and are happy to provide immediate clarification, which is incorporated into the revised manuscript.\
> Theorem 3.2 establishes a worst-case upper bound that provides a fundamental theoretical guarantee on error stability; it is a universal stability guarantee that holds for any valid compression size $r \ge 1$. Concretely, we have $\sigma_{max}(\tilde{I}^T \tilde{I}) = 1$ in the proof of theorem 3.2, the spectral bound remains stable even in extreme compression scenarios. While this theoretical bound is agnostic to $r$, we discuss the practical impacts of the drop rate ($r$) in Section 5, showing that performance improves as $r$ increases. We explicitly added text following Theorem 3.2 to clarify the distinction between this worst-case theoretical guarantee and the empirical behavior, and we mark this as blue.

---

### Review · Reviewer_9ggR · 2025-11-27

**Summary Of Contributions:**

In this paper the author present a method for reducing the cost of computing the Influence function (IF). The influence function can be potentially used in either dataset design or in network understanding by showing the impact of individual data samples on the converge of learning and the network performance. The authors introduce the dropout approach, that is instead of computing the IF in order to reduce the number of parameters (coefficients) that needs to be evaluated. The paper is well written and it seems that it is theoretically correct.

The authors explicitly show that using dropout the error rate does not changes drastically during the computation of the IF it also allows to simplify the gradients compression by reducing the complexity of the projections of the gradients onto a sparser low-dimensional subspace (eq 2 vs eq. 4)


However my main concern is about the fact that while the dropout effectively set some parameters to 0 and reduces the computation, the matrices used for computation remains of the same size. Thus unless one uses specific operators for sparse matrices the gain can be minimal at best. In addition, the paper aims at the efficient computation but the authors did not evaluate the time of the computation which should reflect the efficiency of the method

**Audience:**

Yes

**Audience Explanation:**

The IF function is an important tool in the analysis of the ANNs in general. With the advent of the explainability and the need for understanding in the decision process of the Deep Models this work has a merit to be published.

**Claims And Evidence:**

Yes

**Claims Explanation:**

The authors main claim is that using the dropout reduces the cost of the computation of the IF. By the definition of the IF computation, this is a natural consequence, because the authors statistically remove a selected number of parameters used in gradient computation and thus affecting the overall cost of the IF evaluation.

**Requested Changes:**

Time evaluation of the proposed method with the analysis of the impact of the dropout on the speed of IF computation. This should be contrasted with the IF without the drop out.

---

> ### Author Response · Authors · 2025-12-10
>
> We would like to thank you for your thoughtful review. Below, we address your main concern regarding computational implementation and the time evaluation.
> 1. Sparse matrix operators\
> We appreciate the reviewer raising the point regarding sparse matrix operations. We wish to clarify our implementation details to resolve this misunderstanding. In our method (Section 3.1), we do not simply apply a mask to the full-sized gradient vector (which will indeed retain ($O(d)$) memory cost without sparse formats). Instead, we randomly sample indices and only store and use the values at those $r$ indices. Mechanically, this treats the compressed gradient as a dense vector of size $r$, rather than a sparse vector of size $d$. This ensures that our memory and computational complexity is strictly $O(r)$, rather than $O(d)$, avoiding the overhead of sparse matrix operators entirely.
>
> 2. Time evaluation\
> Regarding the request of time evaluations: We actually provided detailed runtime comparisons in Appendix A.5.2. Table 10 shows that our method (approximate 4.8 s) is significantly faster than DataInf (13.38s) and LiSSA (84.68s) for mislabeled data detection. Table 11 highlights that for the compression process specifically, Dropout takes 9.76s compared to 453.87s for FJLT on GPT2.

---

### Decision · Action_Editor_6orC · 2026-01-13

**Recommendation:** Accept with minor revision

**Additional Comments:**

- The authors may reconsider moving the theorems to the appendix, since these worst case bounds appear to be of little importance (especially when compared to Section 5).
- There is probably a typesetting error: "Note that mH is not invertible..."
- Space is missing in "compression(Choe et al."
- Please ensure that "Gaussian" is consistently written in upper case.
- Consider using \eqref when referring to equations.

**Audience:**

Yes

**Audience Explanation:**

Yes. Influence functions are relevant for explainable and trustworthy models, and could be applied to anomaly detection and for data cleaning. The reviewers agreed that the contribution is interesting for the audience of TMLR.

**Claims And Evidence:**

Yes

**Claims Explanation:**

Yes. All reviewers agreed that after the revision (and including the material presented in the appendices) there is substantial evidence backing the manuscript's claims. There are some reservations regarding the utility of the theorems (which I share), but the empirical evidence seems sound.